# Prompting to Adapt Foundational Segmentation Models

## ABSTRACT

Foundational segmentation models, predominantly trained on scenes typical of natural environments, struggle to generalize across varied image domains. Traditional "training-to-adapt" methods rely heavily on extensive data retraining and model architectures modifications. This significantly limits the models' generalization capabilities and efficiency in deployment. In this study, we propose a novel adaptation paradigm, termed "prompting-to-adapt", to tackle the above issue by introducing an innovative image prompter. This prompter generates domain-specific prompts through few-shot image-mask pairs, incorporating diverse image processing techniques to enhance adaptability. To tackle the inherent non-differentiability of image prompts, we further devise an information-estimation-based gradient descent strategy that leverages the information entropy of image processing combinations to optimize the prompter, ensuring effective adaptation. Through extensive experiments across nine datasets spanning seven image domains (i.e., depth, thermal, camouflage, endoscopic, ultrasound, grayscale, and natural) and four scenarios (i.e., common scenes, camouflage objects, medical images, and industrial data), we demonstrate that our approach significant improves the foundational models' adaptation capabilities. Moreover, the interpretability of the generated prompts provides insightful revelations into their image processing mechanisms. Our source code will be publicly available to foster further innovation and exploration in this field.

## CCS CONCEPTS

• **Computing methodologies** → **Image segmentation**.

## KEYWORDS

Foundational Segmentation Models, Prompt Engineering, Domain Adaption

**ACM Reference Format:**

Anonymous Submission. 2018. Prompting to Adapt Foundational Segmentation Models. In *Proceedings of Make sure to enter the correct conference title from your rights confirmation emai (Conference acronym 'XX)*. ACM, New York, NY, USA, 9 pages. https://doi.org/XXXXXXX.XXXXXXX

## 1 INTRODUCTION

Image segmentation plays a pivotal role in multimedia by facilitating the identification and extraction of information from multi-domain images. This capability is crucial across a broad spectrum of applications, from enhancing autonomous vehicle navigation

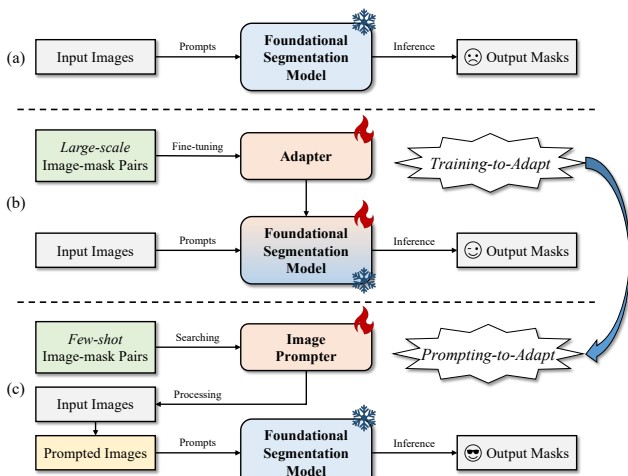

**Figure 1: (a) Foundational segmentation models are typically trained on natural images, encountering issues when applied to images from other domains. (b) "Training-to-adapt" methods generally fit the models through adapters and retraining. (c) Our proposed "prompting-to-adapt" paradigm searches for the optimal combination of image processing prompts in a few-shot manner, adapting the models by altering the context of input images.**

to advancing diagnostic processes in medical imaging. Recent advancements have led to the development of robust foundational models for image segmentation, such as the segment anything model (SAM) [16] and the segment everything everywhere model (SEEM) [44] These models are distinguished by their use of sophisticated prompts ranging from semantic texts that provide contextual understanding to spatial cues that highlight specific areas of interest, which facilitates the accurate segmentation of objects within a diverse array of images. Despite these significant strides, the application of foundational segmentation models to images outside the scope of standard natural scenes presents substantial challenges [3, 36]. This limitation is especially pronounced in domains featuring highly variable conditions, such as medical imagery, industrial scenes, and camouflaged environments. These contexts often exhibit complex background textures, fluctuating lighting conditions, and partial object occlusions, all of which demand an enhanced level of adaptability and precision from segmentation models. Previous methods [3, 36] for adapting foundational segmentation models typically involve intensive training and architectural modifications, which, while effective to a degree, significantly impede the models' ability to generalize and operate efficiently across different domains.

In this paper, we propose an novel adaptation paradigm, changing the perspective from "training-to-adapt" to "prompting-to-adapt" (as illustrated in Fig. 1), to overcome the above challenges. Central to this approach is the development of an innovative image

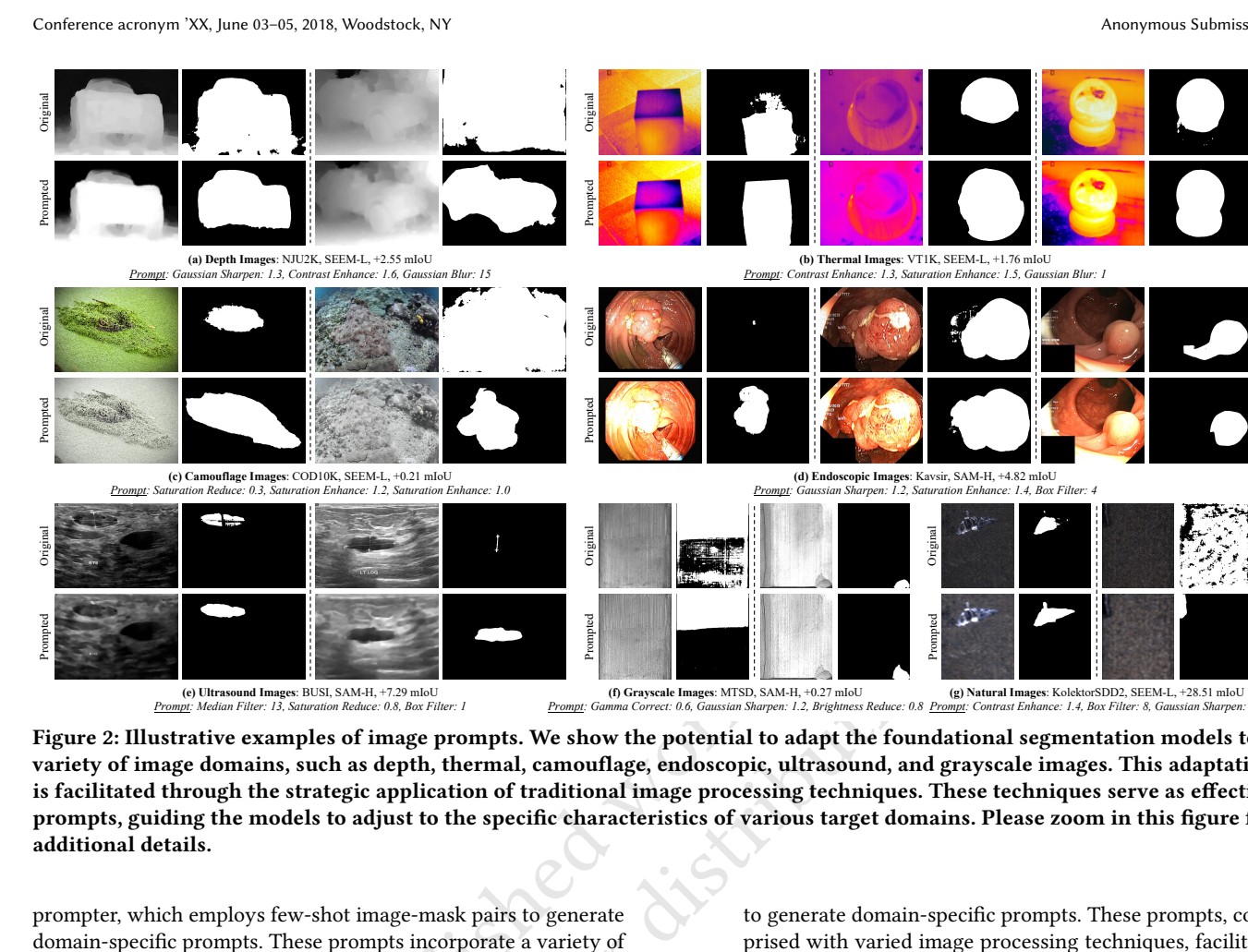

**Figure 2: Illustrative examples of image prompts. We show the potential to adapt the foundational segmentation models to a variety of image domains, such as depth, thermal, camouflage, endoscopic, ultrasound, and grayscale images. This adaptation is facilitated through the strategic application of traditional image processing techniques. These techniques serve as effective prompts, guiding the models to adjust to the specific characteristics of various target domains. Please zoom in this figure for additional details.**

prompter, which employs few-shot image-mask pairs to generate domain-specific prompts. These prompts incorporate a variety of image processing techniques, aiming to adjust the content of input images for the adaptability of segmentation models without necessitating comprehensive model overhauls or extensive fine-tuning. To overcome the intrinsic challenge of non-differentiable image prompts, we further devise an information-estimation-based gradient descent strategy. This strategy capitalizes on the information entropy of image processing combinations, enabling the efficient and effective optimization of the image prompter. Through extensive experiments across diverse domains, we demonstrate the profound impact of our "prompting to adapt" paradigm. This approach not only facilitates significant improvements in the adaptability of foundational segmentation models but also illuminates the interpretability and practical applicability of the generated prompts. For example, as shown in Fig. 2(d), by using the generated prompts, we can modify the lighting conditions of endoscopic images to mitigate the impact of bright spots in original input images.

Our contributions can be summarized as follows:

- We propose a novel adaption paradigm termed "prompting-to-adapt", which significantly enhances the flexibility of foundational segmentation models across a diverse array of image domains. This approach is notable for its departure from traditional, labor-intensive adaptation methods, instead relying on the strategic use of image-mask pairs

to generate domain-specific prompts. These prompts, comprised with varied image processing techniques, facilitate model adaptation without necessitating modifications to model architectures or extensive retraining.

- We propose an information-estimation-based gradient descent strategy that effectively addresses the challenge of non-differentiable image prompts by leveraging the information entropy inherent in the combinations of image processing techniques. Beyond the quantitative enhancements in model performance, we also show the interpretability of the generated prompts. These prompts not only aid in model adaptation but also offer valuable insights into the underlying image processing mechanisms, providing a deeper understanding of how segmentation models can be effectively adapted to new and challenging environments.

- We conduct extensive experiments on *nine* datasets, *i.e.*, NJU2K [15], VT1K [33], CAMO [18], COD10K [7], NC4K [26], Kavsir [12], BUSI [1], MTSD [11], and KolekorSDD2 [2], which cover *seven* image domains including depth, thermal, camouflage, endoscopic, ultrasound, grayscale, and natural images, across *four* scenarios: common scenes, camouflage objects, medical images, and industrial data. By applying our image prompting approach to foundational segmentation models such as SAM [16] and SEEM [44], we demonstrate significant improvements in segmentation performance across various image domains.

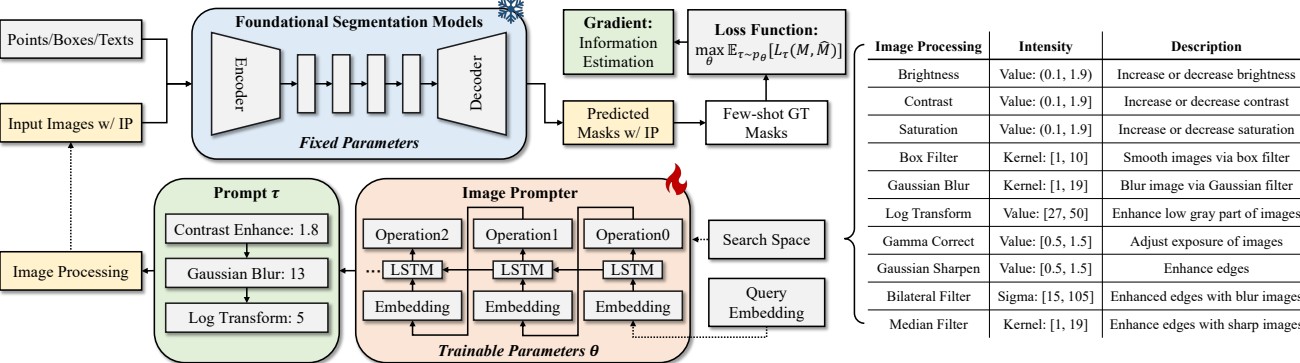

| Image Processing | Intensity | Description |
|---|---|---|
| Brightness | Value: (0.1, 1.9) | Increase or decrease brightness |
| Contrast | Value: (0.1, 1.9) | Increase or decrease contrast |
| Saturation | Value: (0.1, 1.9) | Increase or decrease saturation |
| Box Filter | Kernel: [1, 10] | Smooth images via box filter |
| Gaussian Blur | Kernel: [1, 19] | Blur image via Gaussian filter |
| Log Transform | Value: [27, 50] | Enhance low gray part of images |
| Gamma Correct | Value: [0.5, 1.5] | Adjust exposure of images |
| Gaussian Sharpen | Value: [0.5, 1.5] | Enhance edges |
| Bilateral Filter | Sigma: [15, 105] | Enhanced edges with blur images |
| Median Filter | Kernel: [1, 19] | Enhance edges with sharp images |

**Figure 3: Prompting to adapt foundational segmentation models. During the training process, the segmentation models themselves are kept unchanged. The key to adaptation lies within an image prompter, meticulously trained to craft effective prompts for adaptation. This method is designed to safeguard the core architecture of the segmentation models, thereby maintaining their inherent, pre-trained capabilities. In the inference stage, the generated prompts, which embody a blend of image processing techniques, are directly applied to the input images. This ensures that the foundational segmentation models can adeptly adjust to new inputs, leveraging the specifically designed prompts to achieve enhanced performance without altering the structural essence of the models.**

## 2 RELATED WORK

**Foundational Segmentation Models.** Foundational segmentation models have marked a significant advancement, showcasing remarkable capabilities in accurately segmenting diverse objects. These models, such as the segment anything model (SAM) [16] and the segment everything everywhere model (SEEM) [43, 44], stand out for their utilization of handcrafted prompts to generate detailed segmentation masks. By integrating various types of prompts, including points, boxes, or textual information, these models have proven adept at identifying and segmenting objects across a myriad of images. Despite these advancements, the potential of using the input images themselves as adaptive prompts represents an intriguing avenue for enhancing model adaptability. Our study delves into this concept, exploring the development and application of effective image prompts tailored for foundational segmentation models. Central to our approach is the employment of a few-shot learning strategy aimed at facilitating the models' seamless adaptation to a diverse range of image domains. This strategy underscores our effort to broaden the applicability and versatility of foundational segmentation models, enabling their tailored use in specific tasks or domains that extend beyond their initial training distribution.

**Prompt Engineering.** Prompt engineering has emerged as a strategic approach to repurpose foundation models for novel downstream tasks without necessitating model fine-tuning or modifications. This technique has seen increasing exploration within both the neural language processing and visual foundation model domains, aiming at the automatic derivation of optimal templates for task adaptation [9, 23]. In the context of neural language processing, the methodology for automatic template learning bifurcates into offline and continuous strategies. Offline methods include prompt mining or prompt paraphrasing [13], gradient-based search [31], prompt generation [8], and prompt scoring [5]. On the other hand, continuous methods primarily consist of prefix tuning [20], tuning initialized with discrete prompts [40], hard-soft prompt hybrid tuning [25], and P-Tuning [24]. Similarly, for visual foundation models,

several prompt engineering methods have been proposed. These methods include continuous text token optimization, conditional text token optimization [41], and other approaches [14, 28, 38, 42]. In the segmentation foundation models sector, an approach that parallels our own is the introduction of SAMAug [4], a method that leverages visual point augmentation to boost interactive image segmentation capabilities within the SAM framework. This study introduces a tailored prompt engineering strategy specifically for foundational segmentation models. Our methodology distinguishes itself by examining various image processing techniques to craft effective prompts, aiming to enhance the adaptability and performance of these models.

**Test-Time Adaption.** Test-Time Adaptation (TTA) for image recognition has emerged as a pivotal approach to address the challenges posed by distribution shifts between training and testing data [21]. This technique allows for the fine-tuning of pre-trained models during the inference phase, thereby enhancing their robustness and accuracy when confronted with novel data distributions. TTA is particularly useful in scenarios where the test data distribution is unknown or significantly different from the training data distribution, which is common in real-world applications. TTA methods are categorized into several types, including source-free domain adaptation (SFDA) [17, 19, 22], test-time batch adaptation (TTBA) [30, 32, 39], online test-time adaptation (OTTA) [27, 35], and test-time prior adaptation (TTPA) [29]. SFDA focuses on adapting a model to a new domain without access to original training data. TTBA involves adjusting the model based on batches of test data. OTTA adapts the model continuously as each test instance is processed, learning from each instance to improve subsequent predictions. TTPA utilizes prior knowledge or assumptions about the test data to guide the adaptation process, often incorporating external data or constraints to align the model with the expected test distribution. Our method stands apart from conventional TTA techniques by focusing on dynamic, context-aware adjustments tailored for input images at inference. We aim to identify the best combination of

image processing techniques for the inference stage of a pre-trained segmentation foundation model.

## 3 METHOD

### 3.1 Problem Formulation

The core aim of the "prompting-to-adapt" strategy is to discern optimal image prompts that effectively alter input images, thereby facilitating the adaptation of pre-trained foundational segmentation models to atypical domains. These image prompts are crafted through the strategic combination of various image processing techniques, enabling a seamless adaptation without necessitating alterations to the models' parameters or architecture. We conceptualize the problem as identifying image prompts that adeptly modify input images. This modification aims to produce accurate segmentation masks, leveraging few-shot image-mask ground-truth pairs for precision. Fig. 3 illustrates the comprehensive framework of our approach, which integrates three principal components: a search space, an image prompter, and an information-estimation-based gradient descent strategy. The search space encompasses an extensive array of image processing techniques along with their potential intensities, offering a broad spectrum for prompt construction. Functioning within the defined search space, the image prompter meticulously samples and combines image processing techniques to formulate an image prompt. This prompt specifies the sequence and intensity of the applied image processing techniques. Once the prompt is applied to the input images, generating modified images, these are then utilized in lieu of the original inputs for the foundational segmentation models. The predictions yielded from this process are evaluated against the ground-truth masks using the designated loss function. In the subsequent sections, detailed elucidations of each component integral to the adaptation process will be provided, shedding light on their respective roles and interactions within the overarching framework.

### 3.2 Search Space for Image Processes

In formulating the search space for image prompts, a meticulous examination of various image processing techniques was undertaken, leading to their categorization into two distinct groups: semantic-destructive and semantic-preserving. Semantic-destructive image processing techniques modify the core semantic content of an image. They include operations such as random cropping, rotation, the introduction of noise, and crop-outs. Although these manipulations alter the image's contextual information, they play a crucial role in testing the robustness of segmentation models to changes in image semantics. In contrast, semantic-preserving techniques adjust the visual representation of an image without impacting its semantic integrity. Such adjustments may involve modifications to the brightness, contrast, or saturation levels, serving to enhance or diminish specific image features without altering the underlying content. Despite the potential of semantic-destructive techniques to alter an image's context significantly, their incorporation alongside semantic-preserving techniques is essential for a holistic exploration within the search space. Thus, the constructed search space encompasses a diverse array of operations, including but not limited to, brightness enhancement, contrast reduction, saturation adjustment, box filtering, Gaussian blur, logarithmic

transformation, gamma correction, Gaussian kernel sharpening, bilateral filtering, median filtering, and random rotation. To facilitate the search for optimal operation intensities, we have quantified the intensity range for each operation, dividing it into ten uniformly spaced intervals. This discretization enables a structured approach to exploring the search space, ensuring a comprehensive assessment of various image processing combinations and their effects on the adaptability and performance of segmentation models.

### 3.3 Design of Image Prompter

Within the defined search space, we have developed an image prompter tasked with identifying the optimal combinations of image processing techniques to effectively adapt input images for segmentation tasks. The image prompter is designed to sequence and intensity of the selected image processing operations. Initially, a randomly-initialized query embedding, denoted as $r_0$, is input into a long short-term memory (LSTM) layer [10], producing an operation embedding $r_1$. This embedding subsequently passes through a multi-layer perceptron (MLP) layer, yielding the probability $p(a_1|r_0)$ of selecting a specific image processing technique and its corresponding intensity. A softmax activation function is applied to normalize this probability distribution. The process iteratively continues, leveraging the operation embedding to sequentially generate a comprehensive prompt $\tau_t$, incorporating $t = 1, ..., T$ image processing techniques. While the LSTM layer serves as an illustrative example in our architecture, alternative structures, such as transformers [34], can also be employed to enhance model design flexibility. Parameterizing the image prompter $p(\cdot)$ with parameters $\theta$, we define the formulated image prompts as a probabilistic combination of image processing techniques:

$$\tau_t = R\big( \arg\max p_\theta(a_t|r_{t-1})\big), \tag{1}$$

where $t = 1, ..., T$ and $R(\cdot)$ assigns the highest probability image processing techniques and their intensities. Subsequently, the original input images $I$ undergo processing as follows:

$$I_p = \tau_T\Big(...\tau_2\big(\tau_1(I)\big)\Big), \tag{2}$$

where $I_p$ represents the processed, or prompted, images. The evaluation of the predicted masks is conducted through a loss function defined as:

$$L_\tau(M, f(I_p)) = \mathbb{E}_{m \sim M, \hat{m} \sim f(I_p)}[IoU(m, \hat{m})], \tag{3}$$

where $f(\cdot)$ denotes the foundational segmentation model, $m \sim M$ refers to the ground-truth masks, and $\hat{m}$ to the predicted masks, and $IoU(\cdot, \cdot)$ quantifies the intersection over union between the predicted and ground-truth masks.

### 3.4 Gradient via Information Estimation

Given the defined loss function as shown in Eq. 3, the conventional approach to optimize the parameters of the image prompter is expressed by:

$$\min_\theta \mathbb{E}_{\tau \sim p_\theta}[L_\tau(M, f(I_p))]. \tag{4}$$

However, the inherently non-differentiable nature of the sampling operation within $\tau \sim p_\theta$ precludes the direct application of the gradient of the loss function for parameter updates in the image

**Algorithm 1** Adaption via Image Prompting

---

1: Randomly initialize image prompter with $\theta$.
2: **Repeat**
3:    Predict prompts $\tau \sim p_\theta(a_t|r_t)$ at $t = 0, ..., T - 1$.
4:    Apply the prompts to images.
5:    Input images into foundational segmentation model.
6:    Obtain predicted masks $\hat{m}$.
7:    Calculate loss in Eq. 3 and estimate gradient via Eq. 8.
8:    Update $\theta$ via Eq. 7.
9: **Until** convergence.

---

prompter. To circumvent this, we employ an information-estimation-based gradient descent strategy for the optimization task, focusing on modulating the information entropy of the output probabilities from the image prompter relative to the magnitude of the loss function. This process is formulated as:

$$
\begin{aligned}
\min_\theta & \mathbb{E}_{\tau \sim p_\theta}[L_\tau(M, f(I_p))] \cdot H(a) \\
&= \min_\theta \mathbb{E}_{\tau \sim p_\theta}[L_\tau(M, f(I_p))] \cdot \mathbb{E}_{a \sim p_\theta}[-\log p_\theta(a|r)] \\
&= \max_\theta \mathbb{E}_{a \sim p_\theta}[L_{\tau \sim a}(M, f(I_p)) \cdot \log p_\theta(a|r)],
\end{aligned} \tag{5}
$$

where $H(a)$ represents the information entropy of the output probabilities for the selected combinations of image processing techniques. The rationale is that a lower loss indicates that the prompted images can predict precise masks, suggesting a need to decrease the entropy of the prompt probabilities to diminish the randomness in selecting prompts. Conversely, a higher loss necessitates increasing the randomness to explore more combinations. Accordingly, the gradient of $\theta$ is computed as:

$$
\begin{aligned}
& \nabla_\theta \mathbb{E}_{a \sim p_\theta}[L_{\tau \sim a}(M, f(I_p)) \cdot \log p_\theta(a|r)] \\
& = \mathbb{E}_{a \sim p_\theta}\Big[L_{\tau \sim a}(M, f(I_p)) \cdot \nabla_\theta\big(\log p_\theta(a|r)\big)\Big].
\end{aligned} \tag{6}
$$

This enables updating the parameter $\theta$ via gradient descent:

$$
\theta \leftarrow \theta - \alpha \cdot \nabla_\theta \mathbb{E}_{a \sim p_\theta}[L_{\tau \sim a}(M, f(I_p)) \cdot \log p_\theta(a|r)], \tag{7}
$$

where $\alpha$ is the learning rate. In practice, we discover that the gradient stipulated in Eq. 6 can be directly derived from the gradient of Eq.3, elucidated as follows:

$$
\begin{aligned}
& \mathbb{E}_{a \sim p_\theta}\Big[L_{\tau \sim a}(M, f(I_p)) \cdot \nabla_\theta\big(\log p_\theta(a|r)\big)\Big] \\
& = \int p_\theta(a|r) \nabla_\theta \log p_\theta(a|r) L_{\tau \sim a}(M, f(I_p)) da \\
& = \int \frac{p_\theta(a|r)}{p_\theta(a|r)} \nabla_\theta p_\theta(a|r) L_{\tau \sim a}(M, f(I_p)) da \\
& = \nabla_\theta \int p_\theta(a|r) L_{\tau \sim a}(M, f(I_p)) da \\
& = \nabla_\theta \mathbb{E}_{\tau \sim p_\theta}[L_\tau(M, f(I_p))].
\end{aligned} \tag{8}
$$

This revelation allows for the direct use of the gradient from Eq. 3 to update the image prompter effectively. Overall, the algorithm for adapting foundational segmentation models with image prompter is summarized in Alg. 1.

# 4 EXPERIMENTS

## 4.1 Datasets

To evaluate the adaptability of our proposed method, we conducted a series of rigorous experiments across four categories: common scenes, camouflage objects, medical images, and industrial data, encompassing seven image modalities: depth, thermal, camouflage, endoscopic, ultrasound, grayscale, and natural images.

**Common Scenes.** We utilized the NJU2K dataset [15] for depth imagery, featuring 2,000 stereo images with matching depth maps. Additionally, the VT1K dataset [33] provided 1,000 spatially aligned RGB and thermal infrared image pairs with ground truth annotations. The depth and thermal images from these datasets were directly used as input images for evaluating the adaptability.

**Camouflage Objects.** Three datasets were used, namely the CAMO dataset [18] with 1,250 images, the COD10K dataset [7] with 3,040 training and 2,026 testing images, and the NC4K dataset [26] with 4,121 images, capturing objects in complex, camouflaged settings.

**Medical Images.** The Kavsir dataset [12] provided 1,000 annotated endoscopic images for polyp segmentation, while the BUSI dataset [1] included 780 ultrasound images of breast tumors with detailed annotations.

**Industrial Data.** The MTSD dataset [11] contained 388 magnetic tile images with defect labels, and the KoletorSDD2 dataset [2] included 356 images of surface defects with various imperfections.

## 4.2 Implementation Details

**Search Spaces.** The image processing techniques are systematically categorized into distinct subsets based on image intensity characteristics, as delineated in Fig. 3. For methodologies pertinent to the manipulation of brightness, contrast, and saturation, their respective value spectra were bifurcated into two discrete segments: the first ranging from min value to half of the max value, designated for diminishing effects, and the second from half of the max value to max value, allocated for augmentative adjustments. Consequently, these methods can be denoted using an abbreviation format, where the abbreviation consists of an "id" subscript indicating the numerical values employed. The specific abbreviations are as follows: BE/BR for Brightness Enhance/Reduce, CE/CR for Contrast Enhance/Reduce, SE/SR for Saturation Enhance/Reduce, BX for Box Filter, GB for Gaussian Blur, LT for Log Transform, GC for Gamma Correction, GS for Gaussian Shapes, BF for Bilateral Filter, and MF for Median Filter. All continuous values falling within these defined intervals were meticulously discretized into ten equidistant segments. This granular discretization facilitates a comprehensive exploration of intensity parameters, thereby optimizing the selection of image processing methodologies.

**Search Details.** For each dataset, we sampled 5-shot, 10-shot, and 10%-shot of annotated data from training set for searching and performed validation on the test/validation set. We utilized the SGD optimizer to update the parameter $\theta$, with a momentum value set to 0.9 and a learning rate of 3.5e-4. Training was conducted for 500 epochs. With one A100 GPU, it took approximately 1 hour on average to find a reasonable policy for each dataset.

**Evaluation Metrics.** To rigorously appraise the performance of our proposed method, we adhere to the employment of the mean Intersection over Union (mIoU) as the evaluative metric for the

| Models | Backbone | Common Scenes | | Camouflage Objects | | | Medical Images | | Industrial Data | |
|---|---|---|---|---|---|---|---|---|---|---|
| | | NJU2K | VT1K | CAMO | COD10K | NC4K | Kavsir | BUSI | MTSD | KoletorSDD2 |
| SAM [16] | ViT-B [6] | 55.83 | 57.38 | 37.92 | 49.78 | 49.09 | 62.13 | 52.11 | 54.84 | 46.57 |
| SAM-FT | ViT-B [6] | 57.23 | 57.64 | 38.40 | 49.83 | 49.63 | 60.95 | 56.70 | 54.91 | 52.90 |
| SAM-IP, *ours* | ViT-B [6] | **60.89** | **58.98** | **39.41** | **50.74** | **50.41** | **62.43** | **61.28** | **55.30** | **56.55** |
| Learned Image Prompts, *ours* | GB:0-CE:9-LT:3 | CR:3-CR:0-BR:9 | CR:7-SE:6-SE:1 | BR:4-CE:2-CE:1 | SE:2-CR:0-BR:7 | MF:9-MF:7-GB:9 | MF:4-BX:9-BI:8 | GC:0-BR:1-SR:6 | CE:7-GS:1-GB:7 |
| SAM [16] | ViT-H [6] | 61.37 | 56.58 | 49.00 | 58.39 | 57.23 | 63.29 | 54.54 | 59.24 | 48.04 |
| SAM-FT | ViT-H [6] | 63.47 | 57.44 | 49.43 | 58.50 | 57.45 | 63.81 | 56.59 | 59.31 | 54.24 |
| SAM-IP, *ours* | ViT-H [6] | **65.65** | **60.95** | **50.35** | **58.89** | **58.48** | **64.30** | **60.02** | **59.51** | **60.33** |
| Learned Image Prompts, *ours* | GB:4-CE:8-CR:6 | BE:6-SR:9-MF:4 | BE:0-CR:5-BR:2 | BR:2-SE:2-BR:1 | BE:0-CR:5-BR:2 | LT:0-BI:4-GB:6 | GS:3-MF:2-GB:6 | MF:1-GB:8-SR:8 | GB:2-CE:9-MF:6 |
| SEEM [44] | Focal-T [37] | 57.52 | 65.31 | 55.37 | 54.49 | 60.96 | 38.07 | 43.31 | 38.99 | 33.28 |
| SEEM-FT | Focal-T [37] | 58.93 | 65.69 | 55.49 | 54.57 | 61.10 | 43.21 | 47.97 | 40.27 | 46.11 |
| SEEM-IP, *ours* | Focal-T [37] | **63.56** | **66.20** | **56.21** | **54.91** | **61.29** | **50.20** | **51.75** | **44.33** | **48.78** |
| Learned Image Prompts, *ours* | CE:5-GB:9-GC:7 | GB:0-SR:8-GB:8 | SE:3-SR:4-SE:1 | SE:5-SR:4-SE:4 | CR:2-SE:1-CE:4 | GB:1-CE:8-CE:9 | GB:9-MF:9-BE:9 | CR:5-GB:7-CE:5 | CE:5-MF:0-GB:9 |
| SEEM [44] | Focal-L [37] | 64.48 | 64.61 | 64.17 | 61.45 | 68.60 | 26.15 | 29.58 | 25.54 | 11.58 |
| SEEM-FT | Focal-L [37] | 65.14 | 64.72 | 65.46 | 61.48 | 68.72 | 26.54 | 34.93 | 28.27 | 34.63 |
| SEEM-IP, *ours* | Focal-L [37] | **65.61** | **65.36** | **67.26** | **61.66** | **69.05** | **30.60** | **38.04** | **37.36** | **44.34** |
| Learned Image Prompts, *ours* | SR:8-LT:9-BR:2 | CE:9-SR:9-BE:3 | SR:7-SR:3-GC:0 | SR:7-SE:2-SE:0 | BE:7-SE:7-SR:9 | SE:6-GS:9-LT:8 | GS:7-BI:8-CR:1 | GB:2-CE:9-BI:5 | GS:6-MF:0-GB:8 |

Table 1: Main results of 5-shot learning. 'FT' denotes the employment of fine-tuning techniques with adapter modules, while 'IP' represents the process of evaluating model performance through the application of image prompts during testing.

| Models | Backbone | Common Scenes | | Camouflage Objects | | | Medical Images | | Industrial Data | |
|---|---|---|---|---|---|---|---|---|---|---|
| | | NJU2K | VT1K | CAMO | COD10K | NC4K | Kavsir | BUSI | MTSD | KoletorSDD2 |
| SAM [16] | ViT-B [6] | 55.83 | 57.38 | 37.92 | 49.78 | 49.09 | 62.13 | 52.11 | 54.84 | 46.57 |
| SAM-FT | ViT-B [6] | 60.24 | 58.46 | 38.71 | 49.89 | 49.81 | 64.44 | 57.68 | 55.04 | 53.72 |
| SAM-IP, *ours* | ViT-B [6] | **61.45** | **59.81** | **39.12** | **50.3** | **50.41** | **66.52** | **59.02** | **55.30** | **56.55** |
| Learned Image Prompts, *ours* | BE:5-GC:4-GC:0 | CE:5-GB:8-BR:8 | BR:2-BE:1-CR:4 | BE:0-CR:5-BR:2 | SE:2-CR:0-BR:7 | LT:5-BE:1-GB:6 | GC:9-MF:4-SE:7 | GC:0-BR:1-SR:6 | CE:7-GS:1-GB:7 |
| SAM [16] | ViT-H [6] | 61.37 | 56.58 | 49.00 | 58.39 | 57.23 | 63.29 | 54.54 | 59.24 | 48.04 |
| SAM-FT | ViT-H [6] | 64.26 | 58.67 | 50.43 | 58.61 | 57.64 | 65.21 | 57.29 | 59.51 | 55.67 |
| SAM-IP, *ours* | ViT-H [6] | **65.65** | **60.95** | **50.35** | **59.35** | **58.48** | **66.79** | **60.24** | **59.72** | **57.85** |
| Learned Image Prompts, *ours* | GB:4-CE:8-CR:6 | SR:8-CR:8-GB:9 | BE:0-CR:5-BR:2 | SR:1-GC:1-BX:2 | BE:0-CR:5-BR:2 | CE:7-BX:4-BX:6 | SR:3-GB:5-MF:8 | SE:5-CR:1-GB:0 | CE:7-GS:1-GB:7 |
| SEEM [44] | Focal-T [37] | 57.52 | 65.31 | 55.37 | 54.49 | 60.96 | 38.07 | 43.31 | 38.99 | 33.28 |
| SEEM-FT | Focal-T [37] | 60.16 | 65.82 | 55.49 | 54.62 | 61.14 | 45.62 | 49.89 | 41.64 | 47.19 |
| SEEM-IP, *ours* | Focal-T [37] | **62.55** | **66.20** | **56.21** | **54.97** | **61.29** | **47.10** | **50.75** | **43.38** | **49.17** |
| Learned Image Prompts, *ours* | GB:7-CE:7-SE:4 | GB:0-SR:8-GB:8 | SE:3-SR:4-SE:1 | SE:1-SR:0-GC:6 | CR:2-SE:1-CE:4 | GS:1-SR:9-GC:4 | GC:4-GB:1-GB:1 | SR:0-CE:9-GB:0 | BX:9-CE:6-BX:8 |
| SEEM [44] | Focal-L [37] | 64.48 | 64.61 | 64.17 | 61.45 | 68.60 | 26.15 | 29.58 | 25.54 | 11.58 |
| SEEM-FT | Focal-L [37] | 65.87 | 64.94 | 66.18 | 61.50 | 68.81 | 26.98 | 35.75 | 29.16 | 37.41 |
| SEEM-IP, *ours* | Focal-L [37] | **66.33** | **65.69** | **67.26** | **61.66** | **69.05** | **27.66** | **38.04** | **31.70** | **42.25** |
| Learned Image Prompts, *ours* | BI:2-GC:5-CE:3 | BE:6-GB:9-SR:5 | SR:7-SR:3-GC:0 | SR:7-SE:2-SE:0 | BE:0-SE:4-CE:3 | GS:3-BX:6-MF:0 | GS:7-BI:8-CR:1 | SE:2-BE:9-BX:3 | CE:8-GS:9-BX:8 |

Table 2: Main results of 10-shot learning. 'FT' denotes the employment of fine-tuning techniques with adapter modules, while 'IP' represents the process of evaluating model performance through the application of image prompts during testing.

assessment of segmentation outcomes. The mIoU for an individual image is derived by computing the average Intersection over Union (IoU) score, which is achieved by correlating each predicted mask with its respective ground truth mask. This procedure entails the calculation of the IoU for each predicted mask in relation to all the ground truth masks associated with a given image, culminating in the identification of the highest IoU value. Thereafter, the mIoU is ascertained by averaging these peak IoU values across the entire ensemble of predicted masks present within the image.

## 4.3 Main Results

Tab. 1, Tab. 2, Tab. 3 present the results for the 5-shot, 10-shot, and 10%-shot experimental configurations. Overall, it is observed that the accuracy of both fine-tuning (FT) and the proposed image-prompts (IP) methodology improves with an increase in training samples. Under conditions of limited samples, such as the 5-shot and 10-shot scenarios, fine-tuning exhibits lower performance compared to image-prompts, potentially due to an insufficient sample

| Models | Backbone | Common Scenes | | Camouflage Objects | | | Medical Images | | Industrial Data | |
|---|---|---|---|---|---|---|---|---|---|---|
| | | NJU2K | VT1K | CAMO | COD10K | NC4K | Kavsir | BUSI | MTSD | KoletorSDD2 |
| SAM [16] | ViT-B [6] | 55.83 | 57.38 | 37.92 | 49.78 | 49.09 | 62.13 | 52.11 | 54.84 | 46.57 |
| SAM-FT | ViT-B [6] | 64.55 | 59.54 | **40.05** | 50.63 | 50.24 | 65.47 | 59.47 | **56.72** | 54.32 |
| SAM-IP, *ours* | ViT-B [6] | **65.38** | **59.96** | 39.85 | **51.29** | **50.74** | **66.52** | **61.55** | 55.79 | **56.55** |
| Learned Image Prompts, *ours* | | CE:8-GB:6-LT:5 | GB:4-GS:5-BR:9 | CR:4-GC:0-BX:1 | SR:1-GC1-BX:2 | CR:4-GC:0-BX:1 | GS:2-GC:8-GB:5 | MF:7-GC:9-GC:7 | SR:1-GC:1-BX:2 | CE:7-GS:1-GB:7 |
| SAM [16] | ViT-H [6] | 61.37 | 56.58 | 49.00 | 58.39 | 57.23 | 63.29 | 54.54 | 59.24 | 48.04 |
| SAM-FT | ViT-H [6] | 65.11 | 60.91 | **51.23** | **59.89** | 58.52 | 66.91 | 59.99 | **60.01** | 56.22 |
| SAM-IP, *ours* | ViT-H [6] | **66.18** | **61.78** | 50.99 | 59.35 | **58.98** | **68.11** | **61.83** | 59.51 | **57.85** |
| Learned Image Prompts, *ours* | | GC:7-LT:7-GB:2 | GB:7-BR:9-BI:6 | CR:4-BX:1-BR:1 | SR:1-GC:1-BX:2 | CR:4-GC:0-BX:1 | GS:7-SE:4-BX:3 | MF:6-SR:2-BX:0 | GC:3-GS:7-BR:2 | CE:7-GS:1-GB:7 |
| SEEM [44] | Focal-T [37] | 57.52 | 65.31 | 55.37 | 54.49 | 60.96 | 38.07 | 43.31 | 38.99 | 33.28 |
| SEEM-FT | Focal-T [37] | 61.41 | 66.32 | **56.82** | **55.12** | **61.82** | 49.22 | 50.22 | 43.87 | 48.21 |
| SEEM-IP, *ours* | Focal-T [37] | **62.74** | **66.71** | 56.21 | 54.84 | 61.29 | **51.68** | **52.81** | **45.11** | **49.61** |
| Learned Image Prompts, *ours* | | GB:0-CE:3-CE:2 | SR:2-GC:8-GS:2 | SE:3-SR:4-SE:1 | SE:7-BR:0-CE:0 | CR:2-SE:1-CE:4 | BX:7-CE:3-SE:9 | GB:6-MF:8-BE:7 | GS:1-CE:7-GB:4 | BE:6-BX:9-BI:6 |
| SEEM [44] | Focal-L [37] | 64.48 | 64.61 | 64.17 | 61.45 | 68.60 | 26.15 | 29.58 | 25.54 | 11.58 |
| SEEM-FT | Focal-L [37] | 66.10 | 65.82 | **67.58** | **62.06** | **69.14** | 36.16 | 36.95 | 36.79 | 39.41 |
| SEEM-IP, *ours* | Focal-L [37] | **67.03** | **66.37** | 67.26 | 61.66 | 69.05 | **37.68** | **38.77** | **37.83** | **40.09** |
| Learned Image Prompts, *ours* | | GB:8-CE:6-GB:7 | CE:3-SE:5-GB:0 | SR:7-SR:3-GC:0 | SR:7-SE:2-SE:0 | BE:0-SE:4-CE:3 | SR:7-GC:3-SE:4 | BX:8-SR:3-BX:7 | CE:9-BX:3-SE:6 | CE:4-BX:7-GS:2 |

**Table 3: Main results of 10%-shot learning. 'FT' denotes the employment of fine-tuning techniques with adapter modules, while 'IP' represents the process of evaluating model performance through the application of image prompts during testing.**

| Domain | NJU2K | VT1K | COD10K | Kavsir | BUSI | MTSD | Kolektor |
|---|---|---|---|---|---|---|---|
| NJU2K | +4.81 | +0.21 | -1.06 | -4.52 | -7.97 | -1.51 | +3.16 |
| VT1K | -6.76 | +5.2 | -15.93 | -6.96 | -4.97 | -34.55 | -27.95 |
| COD10K | +1.72 | +0.40 | +1.99 | -1.67 | -3.76 | -3.43 | -2.42 |
| Kavsir | -7.67 | -1.63 | -7.29 | +4.82 | -1.20 | -8.39 | +7.82 |
| BUSI | -1.98 | -0.82 | -15.58 | -3.60 | +4.93 | -27.11 | -10.41 |
| MTSD | -3.46 | +0.25 | -3.95 | -2.93 | -4.31 | +0.27 | -2.21 |
| Kolektor | -1.72 | +0.65 | -8.76 | +3.61 | +3.78 | -11.59 | +9.81 |

**Table 4: Cross-domain transferability of the SAM-H model across diverse datasets.**

| Framework | NJU2K | VT1K | COD10K | Kavsir | BUSI | MTSD | Kolektor |
|---|---|---|---|---|---|---|---|
| H-L | -9.11 | -6.97 | -0.61 | +0.01 | +2.05 | -0.52 | +31.50 |
| L-H | +3.91 | -0.07 | -0.76 | -28.50 | +4.40 | -5.60 | +5.68 |

**Table 5: Cross-framework transferability in foundational segmentation models. The terms H-L and L-H represent the outcomes of transferring policies from the SAM-H model to the SEEM-L framework and vice versa, respectively.**

| Model | NJU2K | VT1K | COD10K | Kavsir | BUSI | MTSD | Kolektor |
|---|---|---|---|---|---|---|---|
| B-H | +4.87 | +4.25 | +1.61 | +3.18 | +7.84 | +0.04 | +9.81 |
| H-B | +5.74 | +2.00 | +1.09 | +3.63 | +3.93 | -2.56 | +9.98 |
| T-L | +0.26 | +0.12 | +0.50 | -0.12 | +1.90 | +6.54 | +27.72 |
| L-T | -0.50 | +0.40 | -1.28 | -11.70 | -2.53 | +1.30 | +14.12 |

**Table 6: Cross-model transferability in foundational segmentation models. The acronyms B-H, H-B, T-L, and L-T correspond to the transfer outcomes from the SAM-B to the SAM-H model, from SAM-H to SAM-B, from SEEM-T to SEEM-L, and from SEEM-L to SEEM-T, respectively.**

size to effectively train fine-tuning adapters. In contrast, image-prompts can still achieve commendable image pre-processing results. Regarding depth images, the analysis indicates a propensity for contrast enhancement operations to be prevalent in the segmentation process of depth images. Visualizations presented in Fig. 2(a) demonstrate significant enhancements in the delineation of foreground and background elements within depth maps as a result of contrast adjustments. In the context of thermal images, the Gaussian blur operation is frequently applied, suggesting that there is a discernible advantage to moderating edge intensity in thermal imagery. Fig. 2(b) visually illustrates the reduction in segmentation noise and the achievement of more refined segmentation

through the employment of image prompts. For camouflage images, although enhancements were observed across various datasets and models as a result of image prompting, the extent of improvement was somewhat limited. This constraint is attributed to the high variability present within camouflage datasets, which presents challenges in identifying a uniform prompt applicable to all scenes. As depicted in Fig. 2(c), the search methodology predominantly concentrates on adjusting color saturation to counteract the effects of color diversity, given the heterogeneous nature of the scenes and the color-dependent distinction between foreground and background elements in camouflage images. In the case of endoscopic images, the findings underscore the recognition and application of processing techniques such as blurring operations (*e.g.*, Gaussian blur, box filter) and color adjustments (*e.g.*, saturation, contrast) to alleviate the impact of illumination-induced bright spots within endoscopic imagery, as exemplified in Fig. 3(d). For ultrasound images, the results indicate a widespread adoption of filtering techniques, including median and box filters, across different models. Moreover, Fig. 2(e) suggests that prompts effectively reduce edge artifacts common in ultrasound imaging, along with the influence

| Data Ratio | 1% | 10% | 20% | 50% | 100% |
|---|---|---|---|---|---|
| # of samples | 8 | 88 | 176 | 440 | 880 |
| mIoU | 65.48 | 68.11 | 68.11 | 67.02 | 67.57 |
| Improvement | +2.19 | +4.82 | +4.82 | + 3.73 | +4.28 |

Table 7: Quantity of training samples. The reference model, designated as SAM-H, achieves a mIoU score of 63.29.

| Sampler | mIoU | Improvement |
|---|---|---|
| Center Points | 68.11 | +4.82 |
| Random Points | 64.14 | +2.33 |
| Everything Mode | 65.99 | +2.70 |

Table 8: Point prompt sampler categories. The baseline model, SAM-H, garners a mIoU of 63.29.

| Number | Policy | mIoU | Improvement |
|---|---|---|---|
| 2 | Contrast Enhance: 1.8
Gaussian Blur: 13 | 67.38 | +4.09 |
| 3 | Gaussian Sharpen: 1.2
Saturation Enhance: 1.4
Box Filter: 4 | 68.11 | +4.82 |
| 4 | Median Filter: 3
Brightness Enhance: 3
Box Filter: 6
Saturation Enhance: 1.3 | 67.85 | +4.56 |

Table 9: Sampled image processing techniques. The baseline model, designated as SAM-H, records a mIoU score of 63.29.

of lesion area annotations, leading to enhanced segmentation results. For grayscale images, the results, in conjunction with Fig. 2(f), demonstrate the application of gamma correction and Gaussian sharpening techniques to bolster edge contrast, thereby augmenting the efficacy of segmentation processes. For natural images with suboptimal imaging quality, the findings, as portrayed in Fig. 2(g), emphasize prompt strategies that primarily concentrate on modulating image contrast or brightness to augment the discernibility of objects targeted for segmentation.

## 4.4 Transferability

This section examines the capacity for image prompts to be transferred across diverse domains, frameworks, and models. The cross-domain experiments, as delineated in Tab. 4, revealed a significant degree of transferability between the Kavsir and KolektorSDD2 datasets. This observation can be ascribed to the common imperative of alleviating the impact of illumination variations present in both datasets. Such a shared necessity highlights the critical role of emphasizing salient objects to augment the precision of segmentation tasks. Moreover, the cross-framework experiments presented in Tab. 5 indicated a constrained transferability of prompts identified by the SAM and SEEM methodologies. This observed limitation is attributed to the variances in the training datasets and the architectural design of the models employed in these two frameworks, implying that these discrepancies present formidable barriers to the transferability of prompts. Finally, the cross-model experiments detailed in Tab. 6) showcased a reciprocal transferability among models within the SAM framework, whereas such transferability was not evident in the SEEM framework. We surmise that the divergent training paradigms among SEEM models may be responsible for this observed inconsistency.

## 4.5 Ablation Study

The ablation studies were meticulously conducted on the Kavsir dataset to dissect and comprehend the various components and their impact on the efficacy of our proposed approach.

**Impact of Searching Sample Quantity.** Tab. 7 elucidates the relationship between the size of the training dataset and the subsequent performance of the image prompts. The study's outcomes underscore the robustness of our methodology, which yields commendable results even under the constraints of a limited training regime, specifically in scenarios where as few as eight images are utilized. This observation is particularly pertinent in scenarios where the availability of annotated data is restricted, thereby underscoring the method's adaptability and resilience.

**Effect of Point Prompt Sampler Diversity.** The results presented in Tab. 8, provide insights into the ramifications of utilizing a variety of point prompt sampling techniques. The empirical evidence

gathered indicates that the adoption of central point forms is associated with the most advantageous outcomes. This finding suggests that the strategic selection of point prompts, particularly those that are centrally located, can significantly enhance the precision and effectiveness of the image segmentation process.

**Influence of Image Processing Method Diversity.** Tab. 9 articulates the benefits of exploring an array of image processing methodologies. The empirical analysis conducted reveals that there exists an optimal threshold in the number of image processing methods that can be effectively integrated into the segmentation algorithm. Specifically, the performance of the algorithm plateaus once the number of explored methods reaches three. This saturation point implies that beyond a certain limit, the incremental addition of image processing techniques does not necessarily translate to a proportional improvement in performance, thereby providing valuable insights into the efficient allocation of computational resources and the strategic selection of image processing methods for optimization purposes.

## 5 CONCLUSION

This paper presents a significant advancement in the domain of image segmentation by introducing the "prompting-to-adapt" paradigm, which addresses the critical issue of limited generalization in foundational segmentation models. Our approach eschews the traditional "training-to-adapt" methods that demand extensive retraining and architectural changes, instead opting for a more efficient and adaptable solution. By employing an image prompter that leverages few-shot learning and diverse image processing techniques, our method significantly improves adaptation capabilities without extensive retraining. Our strategy also introduces an innovative gradient descent method to optimize prompts, ensuring effective domain adaptation. Experiments on nine datasets validate our approach's effectiveness and the interpretability of prompts provides insights into image processing mechanisms.

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
