# OpenReview forum: "Prompting to Adapt Foundational Segmentation Models"
_acmmm.org/ACMMM/2024/Conference — MM2024 Poster_

### Official Review · Reviewer_6nxb · 2024-05-13

**Rating:** 4
**Confidence:** 3

**Summary:**

To address the poor generalization ability of foundational segmentation models on varied image domains, the authors proposed to generate domain-specific prompts to modify the input images. These prompts represent a series of image processing techniques to be applied on the input images, aiming to enhance the model adaptation performance without needing elaborate architecture or extensive re-training. Extensive experimental results across various image domains showed that the proposed method greatly boosts the adaptation performance of foundation segmentation models.

**Strengths:**

- I like the idea of formulating prompts for vision models as a sequence of image processing techniques; it explores a more interpretable prompting technique, and is in general novel and reasonable to me.
- The reported performance gain of the proposed method over fine-tuning with adapters is promising.
- The authors have conducted experiments on a wide range of datasets involving distinct image domains (camouflage, medical, industrial, etc), showing wide applicability of the proposed method.

**Limitations:**

- Details on the experiment
  - The authors didn't seem to disclose the details on the adapter modules they used for they experiments (SAM-FT and SEEM-FT in the tables).
  - There are too few words explaining "cross-domain transferability" experiment in Table 4, making it confusing. Taking Row 3, Column 2 (-6.76) as an example, did the authors train the prompter for SAM-H on VT1K and then directly use the prompter for SAM-H on NJU2K, resulting in a performance drop when compared to the mIOU on VT1K dataset?
  - Adding a comparison between the proposed IP method and full-parameter fine-tuning method can enhance the demonstration of the proposed method's practical applicability. That said, I understand that it may be difficult to fully tune models so large as SAM, so it is totally fine if such comparison is not feasible due to practical factors.

- Discussion on the image prompts
  - This work transforms the input images with image processing prompts. Other works like VPT[1] and EVP[2] that also modulates the inputs to achieves parameter-efficient tuning. What are the merits/insights of learning such transformation on the proposed search space over learning on the pixel space as in [1,2], apart from better interpretability?
- Necessity of "Gradient via Information Estimation"
  - There are a few common-practice tricks, e.g., Gumbel-softmax, that handle the gradient problem when learning undifferentiable processes. Could the authors clarify the advantages of the proposed "gradient via information estimation" method over these existing approaches?

- Insights for specific samples
  - In Fig. 2, some prompts seem to make the target object more difficult to recognize. (e.g., the second grid in Fig. 2(b), the grids in Fig. 2(c), etc.) Would the authors provide some explanation or insights on these counter-factual samples?
  - Also, for some samples, very small difference is obseved between original images and prompted images. However, they lead to significant improvement in the resulted mIOU. So I am curious how sensitive the method is/brings to the perturbations? How much of this sensitivity comes from the foundation model itself?

- Minor
  - Formula. In Eq. (1), the definition of $a$is missing. The definition of $R$and the relationship between $a$and $\tau$, are not easy to follow. In the following text, the mixed use of$a\sim p$, $\tau\sim p$and $\tau\sim a$make it even more confusing. I think it will be much better to clarify these symbols. Also, adding subscripts for $\arg\max$ in Eq. (1) will improve the readability.
  - The designs in "Search Spaces" part (Line 554-560) are quite confusing. Just to confirm: the prompter will maximize the brightness of the image when given an intensity value of 1.0, while minimize it when given intensity value of either 1.0 or 1.9, is that right?

[1]  Visual prompt tuning. (ECCV' 22)

[2] Unleashing the Power of Visual Prompting At the Pixel Level. (TMLR' 24)

**Suitability:**

2

---

### Official Review · Reviewer_NRFW · 2024-05-26

**Rating:** 5
**Confidence:** 3

**Summary:**

This paper introduces “prompting-to-adapt” paradigm in place of the conventional “training-to-adapt” paradigm. The novel paradigm searches for the optimal combination of diverse image processing prompts in a few-shot manner, adapting the models by altering the context of input images. For image prompter, long short-term memory (LSTM) layer is designed to sequence and intensity of the selected image processing operations. To directly use gradient to update the image prompter, this paper further proposes an information-estimation-based gradient descent strategy which leverages the information entropy inherent in the combinations of image processing techniques.

**Strengths:**

1.	The novel “prompting-to-adapt” paradigm provides a new perspective on the adaptation of foundational segmentation models. Better generated prompts can achieve enhanced performance without changing the structure of the model.
2.	The search space comprehensively encompasses a wide variety of semantic destructive and semantic-preserving image processing techniques, ensuring a comprehensive assessment of effects on the adaptability and performance of segmentation models.
3.	 The information-estimation-based gradient descent strategy has been designed to allow for the direct use of the gradient to update the image prompter effectively. This design addresses the challenge of non-differentiable image prompts by leveraging the information entropy inherent in the combinations of image processing techniques.
4.	A comprehensive numerical study has been conducted based on nine benchmarks spanning seven image domains, and results can validate the advantages of the proposed paradigm against the conventional “training-to-adapt” paradigm.

**Limitations:**

1.	The use of LSTM might not be the most efficient approach to design image prompter. Although the paper mentions the use of transformer to design the image prompter, the lack of comparisons with models of other structures leaves room for improvement in the robustness and effectiveness of the methodology.
2.	Tab. 5 shows a constrained transferability of prompts identified by the SAM and SEEM methodologies. While the analysis of the limitation is reasonable, this point proves that the transferability of the paradigm needs to be further improved.
3.	It is not explicitly indicated whether the segmentation results of the original images are obtained by the fine-tuning model or directly from the model without fine-tuning. Visualization of the results obtained in different ways can provide a more intuitive illustration of the effectiveness of the paradigm.

**Suitability:**

2

---

### Official Review · Reviewer_WWFA · 2024-05-31

**Rating:** 4
**Confidence:** 3

**Summary:**

The authors propose a novel adaptation paradigm, termed "prompting-to-adapt", to tackle the limitations of generalization capabilities and efficiency in deployment by introducing an innovative image prompter. First, the image prompter generates domain-specific prompts through few-shot image-mask pairs, incorporating diverse image processing techniques to enhance adaptability. Then, the authors further devise an information-estimation-based gradient descent strategy to tackle the inherent non-differentiability of image prompts. Specifically, this strategy leverages the information entropy of image processing combinations to optimize the prompter, ensuring effective adaptation. Finally, extensive experiments demonstrate that prompting-to-adapt brings significant improvements in segmentation performance across various image domains.

**Strengths:**

1. A key strength of this paper is its novel approach to enhancing the generalizability of foundational segmentation models across diverse image domains through the introduction of the 'prompting-to-adapt' adaptation paradigm.
2. The extensive experimental evaluation on nine datasets spanning four different scenarios robustly demonstrates the superior performance of the proposed method.

**Limitations:**

1. It would be beneficial for the authors to provide a clearer definition and delineation of the task setting to aid the reader's understanding.
2. The comparative performance of the proposed method, as shown in Table 3, indicates a lower performance than the 'FT' method on certain datasets. An in-depth discussion and analysis from the authors regarding this discrepancy would be valuable.
3. Given the current landscape of research, a discussion on how the authors' work relates to and differentiates from existing works, such as [1] and [2], would be advantageous for situating this paper within the field.

[1] Learning to Prompt Segmentation Foundation Models
[2] Customizing Segmentation Foundation Model via Prompt Learning for Instance Segmentation

**Suitability:**

3

---

### Meta-Review · Area_Chair_xg4p · 2024-06-30

**Recommendation:** Accept (Poster)
**Confidence:** 4

**Metareview:**

The manuscript introduces a 'prompting-to-adapt' paradigm designed to enhance the generalization capabilities of foundational segmentation models across various image domains.
Extensive experiments conducted on a diverse array of datasets have demonstrated the method's significant improvements in segmentation performance, underscoring the adaptability and robustness of the proposed approach.
All reviewers commend the novelty of the method and the practicality of the solution.